# Sequencing and Phylogenetic Analysis of the Chloroplast Genome of Three Apricot Species

**DOI:** 10.3390/genes14101959

**Published:** 2023-10-18

**Authors:** Ru Yi, Wenquan Bao, Dun Ao, Yu-e Bai, Lin Wang, Ta-na Wuyun

**Affiliations:** 1College of Forestry, Inner Mongolia Agricultural University, Hohhot 010018, China; 15848145145@163.com (R.Y.); bwq@imau.edu.cn (W.B.); baoad1220@163.com (D.A.); baiyue@imau.edu.cn (Y.-e.B.); 2State Key Laboratory of Tree Genetics and Breeding, Research Institute of Non-Timber Forestry, Chinese Academy of Forestry, Zhengzhou 450003, China; wanglin1815@163.com; 3Key Laboratory of Non-Timber Forest Germplasm Enhancement & Utilization of National Forestry and Grassland Administration, Research Institute of Non-Timber Forestry, Chinese Academy of Forestry, Zhengzhou 450003, China

**Keywords:** Rosaceae, chloroplast genome, comparative genomics, repetitive sequence, phylogenetic relationship

## Abstract

The production and quality of apricots in China is currently limited by the availability of germplasm resource characterizations, including identification at the species and cultivar level. To help address this issue, the complete chloroplast genomes of *Prunus armeniaca* L., *P. sibirica* L. and kernel consumption apricot were sequenced, characterized, and phylogenetically analyzed. The three chloroplast (cp) genomes ranged from 157,951 to 158,224 bp, and 131 genes were identified, including 86 protein-coding genes, 37 rRNAs, and 8 tRNAs. The GC content ranged from 36.70% to 36.75%. Of the 170 repetitive sequences detected, 42 were shared by all three species, and 53–57 simple sequence repeats were detected with AT base preferences. Comparative genomic analysis revealed high similarity in overall structure and gene content as well as seven variation hotspot regions, including *psbA-trnK-UUU*, *rpoC1-rpoB*, *rpl32-trnL-UAG*, *trnK-rps16*, *ndhG-ndhI*, *ccsA-ndhD*, and *ndhF-trnL*. Phylogenetic analysis showed that the three apricot species clustered into one group, and the genetic relationship between *P. armeniaca* and kernel consumption apricot was the closest. The results of this study provide a theoretical basis for further research on the genetic diversity of apricots and the development and utilization of molecular markers for the genetic engineering and breeding of apricots.

## 1. Introduction

Apricot is a deciduous tree species, which belongs to section Armeniaca (Lam.) Koch genus *Prunus* of Rosaceae family (2n = 16) [1,2]. Ten species of apricot have been identified, and they are widely distributed worldwide, including *P*. *armeniaca*, *P*. *sibirica*, *P*. *mandshurica*, *P*. *zhengheensis*, *P*. *dasycarpa*, *P*. *holosericea*, *P*. *zhidanensis*, *P*. *mume*, *P*. *limeixing*, and *P*. *byigantina*. Four distinct species are most commonly recognized: *P. armeniaca*, *P. mandshurica*, *P. sibirica*, and *P. mume* [3]. Apricot is of Chinese origin and has been cultivated in China for more than 3000 years [4]. This study focused on *P. armeniaca*,*P. sibirica*, and kernel consumption apricots which are widely distributed in northern China [5]. The fruits of *P. armeniaca* are fresh with unique aroma, delicious taste, but also contains a variety of organic components, vitamins and inorganic salts, high nutritional value, wide range of uses, and can be processed into dried apricots, cultivated around the world, accounting for a high proportion of global fruit production [6,7]. *P. sibirica* has high ecological value as a pioneer tree species for vegetation restoration because it is cold, drought, and poor-soil tolerant. In addition, the bitter kernel of *P. sibirica* has an amygdalin content of 3.5–7.6%, and it is also rich in vitamins, selenium, calcium, phosphorus, iron, potassium and other nutrients. Thus, this kernel is a raw material in traditional Chinese medicine [8]. The kernel consumption apricots have typical characteristic of large and sweet kernels. And the kernels have a crude fat content of ~60% and thus can be used to produce kernel oil, and they have a protein content of ~30%, including eight kinds of essential amino acids for the human body. Thus, kernels are a high-quality plant protein raw material [9].

Chloroplasts are unique endosymbiotic organelles found in plants and photosynthetic algae [10], serving as the primary site of photosynthesis and supplying energy for plant growth and development and carbon intermediates for a number of critical metabolic reactions. In addition, chloroplasts play an important role in plant response to light, heat, drought, salt, and other stresses [11]. The chloroplast (cp) genome is maternally inherited in most angiosperms or paternally inherited in some gymnosperms. The sequence analysis of the double-stranded circular DNA cp genome is important in various areas of study, including the development of linked molecular markers, the reconstruction of phylogenetic relationships, and the genetic engineering and breeding of plants [12]. The cp genome exhibits a highly conserved organization composed of a pair of inverted repeats (IRs), a large single-copy (LSC) region, and a small single-copy (SSC) region [13,14]. The two IR regions, which are separated by the LSC and SSC regions, are equal in length and opposite in direction; however, variations have been observed in some plants, mainly presented as IR loss, contraction, expansion, and sequence direction changes [15]. Early research studies on the cp genome focused mainly on understanding the evolutionary history of chloroplasts and safeguarding uncommon and endangered plants [16]. With improvements in sequencing technology, the complete cp genome of *Nicotiana tabacum* was obtained for the first time in 1986 [17]. More recently, the cp genomes of *Prunus cerasus* (sour cherry) [18], *P. phaeosticta* (dark-spotted cherry) [19], *P. kansuensis* (Chinese bush peach) [20], and *P. japonica* (Japanese bush cherry) [21] were sequenced and analyzed, and their phylogenetic positions and genetic relationships were determined.

Morphological characteristic analyses can preliminarily reveal the morphological characteristics and genetic variations of plants [22]. However, the morphological characteristics of apricot are influenced by the environment and gene dominance, and the period required to obtain morphological characteristics is long [23]. With the rapid development of next-generation sequencing technology and phylogenetic genomics, cp genome sequencing has been widely used in molecular evolution and phylogenetic studies of many plant species [24]. More accurate classifications and phylogenetic relationships of apricot can be obtained through the combination of cp genome sequencing and phylogenetic genomics.

In this study, *P. armeniaca*, *P. sibirica*, and kernel consumption apricots were used as the research objects to obtain the cp genome sequences, and then, the sequences were spliced, annotated, and compared. A phylogenetic analysis was performed, and the evolutionary relationship between *P. armeniaca*, *P. sibirica*, and kernel consumption apricot was systematically studied at the cp genome sequence level. The results provide a reference for future taxonomic and phylogenetic analyses and molecular marker development of apricot and a molecular guide for genetic engineering and breeding.

## 2. Materials and Methods

### 2.1. Sample Material Collection, DNA Extraction, and Sequencing

Fresh tender leaves were collected from the cultivated variety *P. armeniaca* in Mentougou, Beijing (Sungold), wild resource of *P. sibirica* in Wanjiagou (F106), Inner Mongolia, and the cultivated variety of kernel consumption apricot growing in Wei County, Hebei (Youyi). The samples were stored at −80 °C.

Total genomic DNA was extracted using a Plant Genomic DNA Kit (Tiangen, Beijing, China). DNA quality and quantity were detected using a NanoDrop 2000 spectrophotometer (Thermo Fisher Scientific, Waltham, MA, USA) and 0.8% agarose gel electrophoresis, and the DNA was further fragmented for sequence library preparation following fragment purification and end repair. After testing the library preparation, DNA sequencing was performed using the Illumina HiSeq X high-throughput platform (Illumina, San Diego, CA, USA). Library preparation and sequencing were performed by BGI Genomics (Shenzhen, China).

### 2.2. Chloroplast Genome Assembly and Annotation

Using SOAPdenovo (http://soap.genomics.org.cn/soapdenovo.html, accessed on 5 September 2016), the reads were mapped to the cp genome of *P. persica*, which was downloaded from GenBank (NC_014697.1). Contigs obtained by de novo assembly mapping to the consensus sequence were obtained using the reference genome to check the errors or ambiguities resulting from either assembly method. Gapcloser (https://sourceforge.net/projects/soapdenovo2/files/GapCloser/, accessed on 7 September 2016) was used to modify the spaces between long contigs to obtain a complete cp genome. The three apricot cp genome sequences were preliminarily annotated using DOGMA and CpGAVAS (http://phylocluster, accessed on 11 September 2016), and the annotation was completed by manually modifying the start and stop codons of individual genes. Geneious was used for manual corrections. Finally, the annotated cp genomes of the three apricot species were submitted to GenomeVx (http://wolfe.gen.tcd.ie/GenomeVx/, accessed on 10 March 2023) to complete the physical mapping.

### 2.3. Repeat Sequences and Simple Sequence Repeat (SSR) Analysis

REPuter (http://bibiserv.techfak.uni-biele.org.de/reputer/, accessed on 15 March 2023) was used to predict the scattered repetitive sequences of the cp genomes, including forward (F), reverse (R), complementary (C), and palindromic (P) repeats. The conditions for repeat sequences included a sequence length ≥18 bp and similarity ≥ 90%. Simple sequence repeat loci in the apricot cp genome were detected using MISA (https://webblast.ipk-gatersleben.de/misa/index.php, accessed on 20 March 2023) with the following parameters: ≥6 mononucleotide repeats, ≥4 dinucleotide and trinucleotide repeats, and ≥3 tetranucleotide, pentanucleotide, and hexanucleotide repeats.

### 2.4. Chloroplast Genome Comparison and Analysis of Variations in IR/SC Boundaries

mVISTA (http://genome.lbl.gov/vista/index.shtml, accessed on 25 March 2023) was used to compare similarities and variations in the cp genomes among *P. armeniaca*, *P. sibirica*, kernel consumption apricot, *P. mume*, *P. pyrifolia*, and *N. tabacum*, with the *P. persica* cp genome sequence serving as the reference sequence. Based on the annotation information of the cp genome, the LSC, SSC, and IR boundary sequences in the apricot cp genomes were compared with those in the *P. mume*, *P. persica*, *P. pyrifolia*, and *N. tabacum* cp genomes. The IR-SC boundary of the cp genome was visualized using IRscope.

### 2.5. Phylogenetic Analysis

The cp genome sequences of 27 angiosperms and 2 gymnosperms were selected from the National Center for Biotechnology Information (NCBI) database for phylogenetic analysis (Appendix A). The phylogenetic tree was constructed by extracting 77 common protein-coding gene sequences from the 29 species and using *Pinus thunbergia* (NC_001631) and *Ginkgo biloba* (NC_016986) as outgroups. The 77 common protein coding genes were *atpA*, *ndhA*, *atpB*, *atpE*, *ndhC*, *atpF*, *atpH*, *ndhD*, *atpI*, *ccsA*, *ndhE*, *cemA*, *clpP*, *ndhF*, *infA*, *matK*, *ndhK*, *petA*, *petB*, *ndhG*, *petD*, *petG*, *ndhH*, *petL*, *petN*, *ndhI*, *psaA*, *psaB*, *ndhJ*, *psaC*, *psaI*, *psbA*, *psaJ*, *rpl22*, *psbB*, *rpl23*, *rps16*, *psbC*, *rbcL*, *rpl14*, *psbD*, *rpl16*, *rpl2*, *psbE*, *rpl20*, *rpl32*, *psbF*, *rpl33*, *rpl36*, *psbH*, *rpoA*, *rpoB*, *psbI*, *rpoC1*, *rpoC2*, *psbJ*, *rps11*, *rps12*, *psbK*, *rps14*, *rps15*, *psbL*, *rps18*, *rps19*, *psbM*, *rps2*, *rps3*, *psbN*, *rps4*, *rps7*, *psbT*, *rps8*, *ycf3*, *psbZ*, *ycf2*, and *ycf4*. MEGA was used for protein sequence alignment, and the maximum likelihood (ML) method of analysis was used to construct a phylogenetic tree. Visualization of the system tree was completed using Figtreev1.4.4.

## 3. Results

### 3.1. Organization and Features of the Chloroplast Genomes

The total cp genome lengths of *P. armeniaca*, *P. sibirica*, and kernel consumption apricot were 157,951, 158,224, and 157,994 bp, respectively. The cp genomes showed a typical tetrad structure, including two IRs (26,373 bp) that were separated by LSC (86,217–86,358 bp) and SSC (18,988–19,120 bp) regions. Of the total cp genome, the protein-coding region accounted for (49.63–49.72%); tRNA and rRNA accounted for (1.77–1.78%), and (5.72–5.73%), respectively; meanwhile, introns and intergenic spacers (IGSs) accounted for (11.40–11.41%) and (31.34–31.47%), respectively. The GC content of the cp genomes of *P. armeniaca*, *P. sibirica*, and kernel consumption apricot was 36.74%, 36.70%, and 36.75%, respectively. In addition, the protein-coding region accounted for 37.63–37.64% of the genome; tRNA and rRNA accounted for 53.25–53.35% and 55.50%, respectively; meanwhile, introns and IGSs accounted for 36.61–36.68%, and 30.65–30.83%, respectively. The GC content of the IR region (42.57–42.59%) was higher than that of the LSC region (34.52–34.56%) and SSC region (30.30–30.50%), suggesting the higher stability of the IR region than the LSC and SSC regions (Table 1).

The cp genomes of the three apricot species were assembled and annotated (Figure 1). Functional analyses classified them into four categories: self-replication-related, photosynthesis-related, other, and unknown functions (Table 2). A total of 131 genes were predicted and annotated, including 86 protein-coding genes, 37 tRNAs, and 8 rRNAs (Table 2). Among these, 18 genes (*atpF*, *clpP*, *ndhA*, *ndhB*, *petB*, *petD*, *rpl2*, *rpl16*, *rpoC1*, *rps12*, *rps16*, *trnA-UGC*, *trnG-GCC*, *trnI-GAU*, *trnK-UUU*, *trnL-UAA*, *trnV-UAC*, and *ycf3*) contained one intron, and two genes (*clpP* and *ycf3*) contained two introns (Appendix A).

### 3.2. Characterization of Repeat Sequences and SSRs

There were 170 repeat sequences in the combined apricot cp genomes, including 57 forward repeats (F), 53 palindromic repeats (P), 56 reverse repeats (R), and 4 C complementary repeats (C). Among the apricots, *P. sibirica* had the most repeat sequences (63), followed by kernel consumption apricot (55) and *P. armeniaca* (52) (Figure 2a). Repeat sequence lengths of 18–20 bp (64.71%) and 21–25 bp (25.88%) dominated (Figure 2b). Of the 170 repeats identified, 42 were shared by the three species, while there were 13, 10, and 21 unique repeat sequences obtained for *P. armeniaca*, *P. sibirica*, and kernel consumption apricot, respectively (Figure 2c and Appendix A).

The three apricot cp genomes contained three forms of SSRs: mononucleotide, dinucleotide, and compound (Figure 3a). There were 53 SSRs in *P. sibirica*, 56 in *P. armeniaca*, and 57 in kernel consumption apricot. Mononucleotide repeats (ranging from 84.91% in *P. sibirica* to 94.64% in *P. armeniaca*) occurred most frequently, followed by dinucleotide (ranging from 3.57% in *P. armeniaca* to 10.53% in kernel consumption apricot) and compound SSRs (ranging from 1.76% in *P. armeniaca* to 3.77% in *P. sibirica*). The number of A/T mononucleotide repeats (ranging from 78.95% in kernel consumption apricot to 81.13% in *P. sibirica*) was greater than that of C/G repeats (ranging from 8.77% in kernel consumption apricot to 9.43% in *P. sibirica*). The quantity of dinucleotide repeats, including AT/TA repeats, ranged from 5.66% in *P. sibirica* to 8.93% in *P. armeniaca* (Appendix A). We further analyzed SSR distribution and found that most were distributed in the LSC region (83.93–85.96%); far fewer were in the SSC (10.53–12.50%) and IR (3.51–3.77%) regions (Figure 3b and Appendix A).

### 3.3. Comparative Analysis of Apricot Chloroplast Genomes

A comparison of the three apricot cp genomes indicated that the coding region is relatively conserved, with variations mainly occurring in the intergenic and intron regions. Intergenic spacer regions involving the *psbA-trnK-UUU*, *rpoC1-rpoB*, *rpl32-trnL-UAG*, *trnK-rps16*, *ndhG-ndhI*, *ccsA-ndhD*, and *ndhF-trnL* genes are hotspots for apricot cp genome variation (Figure 4). These hotspots can provide vital sequence information for the design of screening DNA barcodes and phylogenetic analyses of apricot species.

### 3.4. Analysis of Variations in the IR/SC Boundaries

We compared the cp genome IR regions of *P. armeniaca*, *P. sibirica*, and kernel consumption apricot with those of *P. mume*, *P. persica*, *P. pyrifolia*, and *N. tabacum* (Figure 5). The *rps19* gene was detected at the IRb/LSC boundary in *P. armeniaca*, *P. sibirica*, kernel consumption apricot, *P. mume*, *P. persica*, and *P. pyrifolia.* The fragment size in the IRb region was 120–197 bp. In contrast, in *N. tabacum*, there was no pseudogene of *rps19* in the IRb/LSC boundary region in *N. tabacum*. The *ycf1* gene was found in the IRa/SSC boundary of *P. armeniaca*, *P. sibirica*, kernel consumption apricot, *P. mume*, *P. persica*, *P. pyrifolia*, and *N. tabacum*. The fragment size of *Ψycf1* in the IRa region was 996–1073 bp. The *Ψycf1* pseudogene in the IRb/SSC region of *P. armeniaca*, *P. sibirica*, kernel consumption apricot, *P. mume*, *P. persica*, and *P. pyrifolia* exhibited different lengths of overlap with that of *ndhF*, whereas *ycf1* of *N. tabacum* did not overlap with that of *ndhF*.

### 3.5. Phylogenetic Analysis

According to the constructed phylogenetic tree, the support rate for each branch was high (>80%), while the support rate for 21 of the 27 nodes was >90% (Figure 6). The 29 species analyzed were divided into seven groups: EUROSIDS I, EUROSIDS II, EUASTERIDS II, EUASTERIDS I, basal angiosperms, monocots, and gymnosperms. Rosales, Cucurbitales, Fabales, and Malpighiales were clustered together to form EUROSIDS I, and other plants were clustered together in turn. Gymnosperms and basal angiosperms were obviously clustered into one branch. In the phylogenetic tree, *P. armeniaca*, *P. sibirica*, and kernel consumption apricot clustered in the Rosales order within EUROSIDS I. The analysis shows that the relationship between *P. armeniaca* and kernel consumption apricot is the closest.

## 4. Discussion

In this study, the cp genomes of three apricot species were successfully sequenced, assembled, analyzed, and compared. The structural characteristics of the three cp genomes were similar, which exhibited typical tetrad structures [25,26]. The total length of the cp genomes ranged from 157,951 to 158,224 bp. The number of genes was consistent, with 131 genes each, including 86 protein-coding genes, 37 tRNAs, and 8 rRNAs, which is consistent with the previously reported cp genomes of *P. mume* [27], *P. persica* [26], and *P. pyrifolia* [28]. The GC content of the three apricot cp genomes was also similar (36.70–36.75%), and the GC content in the IR region was the highest (42.57–42.59%) owing to the GC-rich rRNA and tRNA in the IR region.

Repeat sequences play an important evolutionary role in the cp genome by promoting cp genome rearrangement, inducing genomic structural changes, and increasing population genetic diversity [29,30]. There were 170 repeats in the cp genome, among which, F, R, and C repeats were the main repetitive sequences (97.65%). Different abundances of palindromic repeats in the cp genomes may provide additional evolutionary information, as the presence and abundance of repeats in the cp genome may contain phylogenetic signals [31]. The total number and proportion of repeat types in the three apricot species showed a similar pattern, suggesting a similar evolutionary history and closer affinities among these species. SSRs related to genome rearrangement and recombination are widely distributed in the cp genome, and they are prone to dislocation during DNA replication, which leads to rich polymorphisms that provide information for marker development in population genetics and evolutionary research [24]. SSRs of plant chloroplast genes are mainly mononucleotide and dinucleotide repeats, and trinucleotide to hexanucleotide repeats are relatively less than mononucleotide and dinucleotide repeats [32]. The SSRs found in the three apricot species were dominated by mononucleotide repeats, especially poly (A/T), which is similar to that of other Rosaceae species [33]. The repeat sequences and SSRs detected in this study can provide useful information for future research on the evolution of apricot species.

Previous studies have shown that changes in genome size mainly occur in the SSC and LSC regions and are highly conserved in the IR regions [34,35]. Variation in the non-coding region was significantly higher than that in the coding region owing to the large selection pressure [36]. In particular, the intergenic regions, including *psbA-trnK-UUU*, *rpoC1-rpoB*, *rpl32-trnL-UAG*, *trnK-rps16*, *ndhG-ndhI*, *ccsA-ndhD*, and *ndhF-trnL*, are highly variable regions, which can be used as DNA barcodes for future phylogenetic analyses of apricots.

Contraction and expansion are the main causes of cp genome evolution, mainly occurring at the *rps19*, *ycf1*, *trnH-GUG*, and *ndhF* positions [37,38]. Although the gene distribution of the four main regional boundaries in the three apricot cp genomes showed the same pattern, there were differences in the microstructure, especially the location of *rps19*, *ycf1*, *ndhF*, and *trnH-GUG.* The *rps19* crosses the boundary between the LSC region and the IR region, which is similar to previous observations in *P. mume*, *P. armeniaca*, and *P. salicina* [39]. The differences in the lengths of these four genes in the IR/SC boundary region can be used to identify *P. armeniaca*, *P. sibirica*, and kernel consumption apricot.

The cp genome of angiosperms is maternally inherited and an independent evolutionary system with a moderate rate and can be used for phylogenetic analyses of each classification level [40]. Especially in the study of the phylogenetic relationship between angiosperms and some controversial species, cp genome analysis is the preferred research method [41]. The results of the phylogenetic analysis showed that the 29 species included in the phylogenetic tree could be divided into 7 groups: EUROSIDS I, EUROSIDS II, EUASTERIDS II, EUASTERIDS I, basal angiosperms, monocots, and gymnosperms. The support rate for each branch was high (>80%), in which Rosales, Cucurbitales, Fabales, and Malpighiales were clustered together to form EUROSIDS I, which is consistent with the results of APG III [42]. Phylogenetic tree analysis in our study showed that *P. armeniaca*, *P. sibirica*, and kernel consumption apricot all clustered together, which was consistent with the results of traditional morphological analysis [43] and genetic diversity analysis [44]. This study has certain limitations, and the taxonomic status of apricot needs to be further analyzed to more clearly reveal the taxonomic status and origin of apricot.

## 5. Conclusions

In this study, we sequenced and analyzed the complete cp genome of three apricots. The results showed that the cp genomes showed a typical tetrad structure. Comparative analysis of the cp genomes revealed that the organization and gene order were highly conserved. The cp genome size, GC content, gene number, and gene arrangement order were similar in the three apricot species. We detected abundant long-repeat sequences and SSR loci in the three apricot species. The IR/SC boundary regions were similar but also exhibited some microstructural differences among the three species. Seven significant differences were identified in the non-coding regions of the three cp genomes, which can be exploited in the DNA barcoding of apricot. Finally, phylogenetic tree analysis supported a close relationship among the three apricot species. The results are of great significance for studies on the internal structure of the cp genome of apricots and the breeding, environmental adaptation, and hybrid breeding of apricots.

## Figures and Tables

**Figure 1 genes-14-01959-f001:**
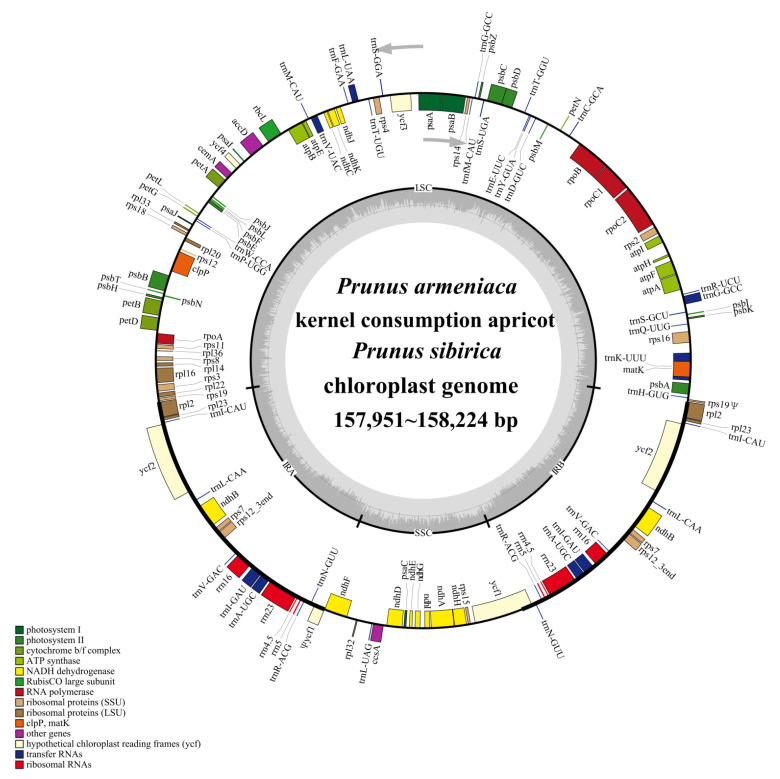
Combined gene map of the chloroplast genome of the three apricot species.

**Figure 2 genes-14-01959-f002:**
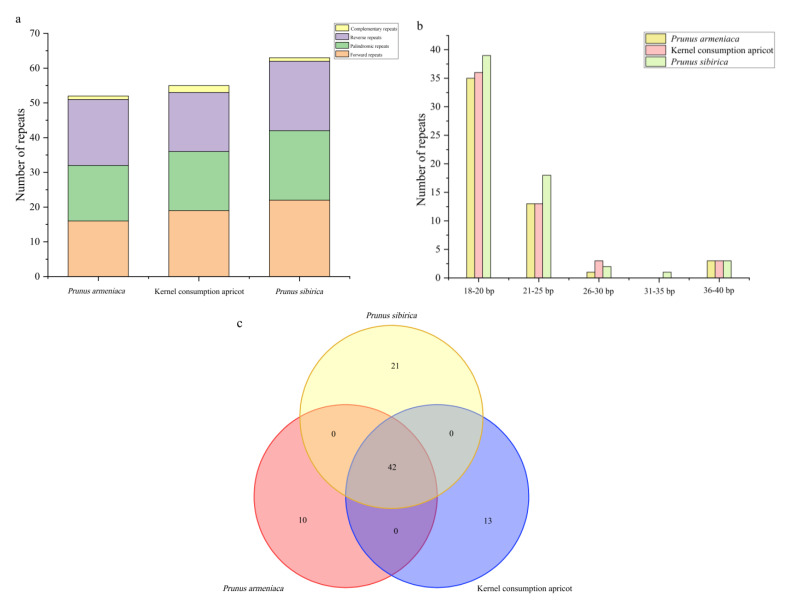
Analysis of repeat sequences in the chloroplast genomes of the three apricot species. (**a**) Frequency of the repeat type. (**b**) Frequency of repeat sequences by length. (**c**) Number of common and unique chloroplast genome repeat sequences.

**Figure 3 genes-14-01959-f003:**
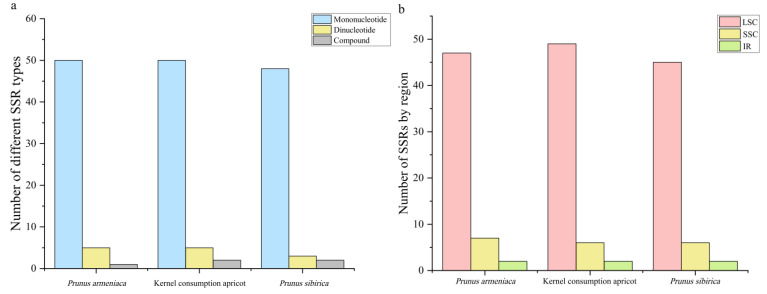
Frequency of SSRs in three apricot species. (**a**) Number of SSRs by type. (**b**) Number of SSRs by genome region.

**Figure 4 genes-14-01959-f004:**
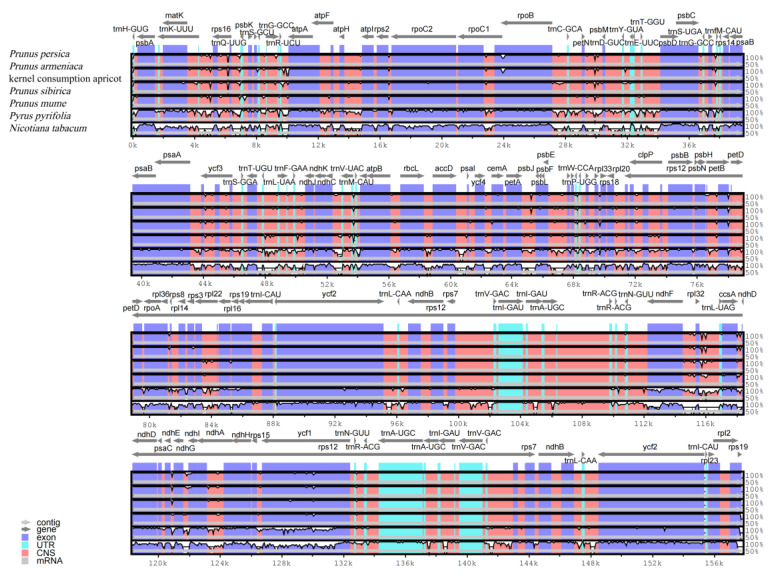
Comparison of the chloroplast genome sequences of *P. armeniaca*, kernel consumption apricot, *P. sibirica*, *P. mume*, *P. pyrifolia*, and *N. tabacum*.

**Figure 5 genes-14-01959-f005:**
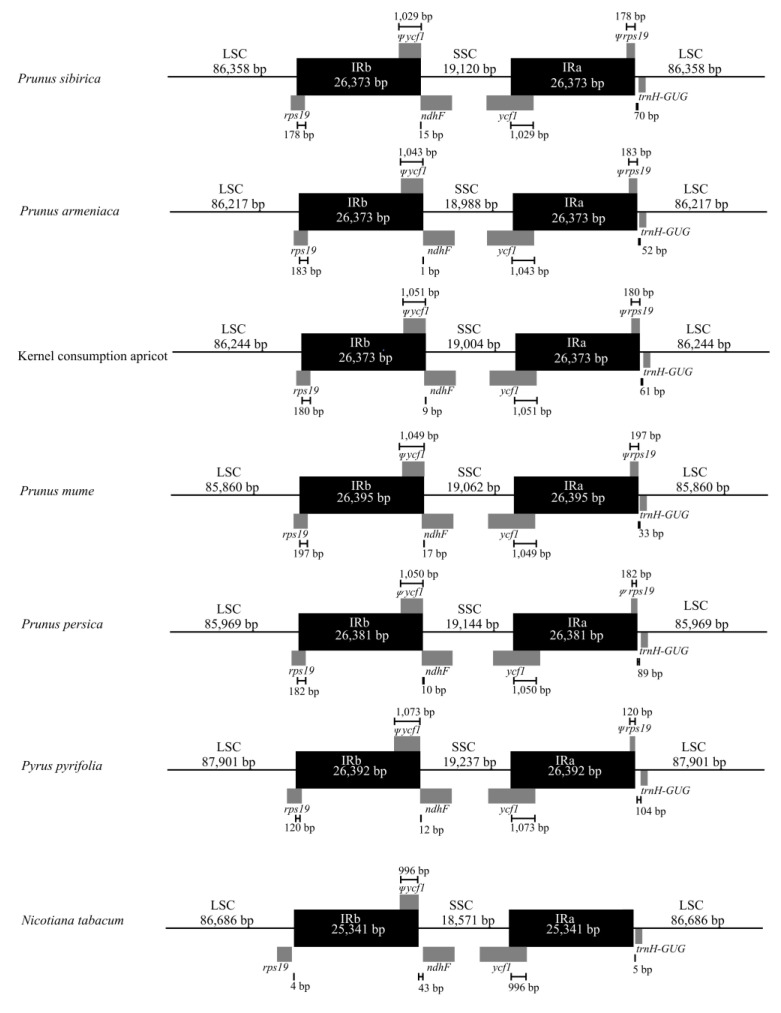
Comparison of the IR-SC region of the chloroplast genome in *P. sibirica*, *P. armeniaca*, kernel consumption apricot, *P. mume*, *P. persica*, *P. pyrifolia*, and *N. tabacum*.

**Figure 6 genes-14-01959-f006:**
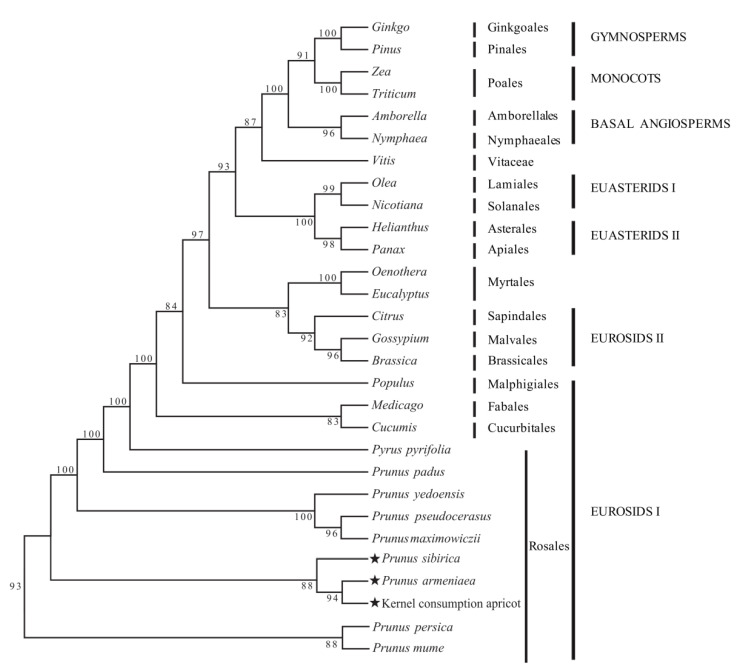
Phylogenetic relationship of the three apricot species reconstructed using the maximum likelihood (ML) method. ★ The star is marked by the study of three apricots.

**Table 1 genes-14-01959-t001:** Chloroplast genome characteristics of three apricot species.

Characteristics	*P. armeniaca*	Kernel Consumption Apricot	*P. sibirica*
Size, bp	157,951	157,994	158,224
LSC ^1^, bp (%)	86,217 (54.58)	86,244 (54.59)	86,358 (54.58)
SSC, bp (%)	18,988 (12.02)	19,004 (12.03)	19,120 (12.09)
IR, bp (%)	26,373 (16.70)	26,373 (16.69)	26,373 (16.67)
Protein-coding regions, bp (%)	78,530 (49.72)	78,532 (49.71)	78,532 (49.63)
Introns, bp (%)	18,017 (11.41)	18,005 (11.40)	18,049 (11.41)
rRNA, bp (%)	9,048 (5.73)	9,048 (5.73)	9,048 (5.72)
tRNA, bp (%)	2,804 (1.78)	2,804 (1.78)	2,806 (1.77)
IGS, bp (%)	49,552 (31.34)	49,605 (31.40)	49,789 (31.47)
Total GC content (%)	36.74	36.75	36.7
LSC GC content (%)	34.55	34.56	34.52
SSC GC content (%)	30.5	30.46	30.3
IR GC content (%)	42.58	42.59	42.57
Protein-coding regions’ GC content (%)	37.63	37.63	37.64
Introns GC content (%)	36.68	36.68	36.61
IGS GC content (%)	30.77	30.83	30.65
rRNA GC content (%)	55.5	55.5	55.5
tRNA GC content (%)	53.35	53.25	53.28
Total number of genes	131	131	131
Protein-coding genes	86	86	86
tRNA genes	37	37	37
rRNA genes	8	8	8
GenBank accession numbers	KY101151	KY101150	KY101154

^1^ Abbreviations: LSC, large single-copy; SSC, small single-copy; IR, inverted repeat; IGS, intergenic spacer.

**Table 2 genes-14-01959-t002:** Genes present in the chloroplast genomes of the three apricot species.

Gene Categories	Gene Groups	Gene Names
Self-replication	Ribosomal RNAs	*rrn16* ^b^, *rrn23* ^b^, *rrn4.5* ^b^, *rrn5* ^b^
	Transfer RNAs	*trnA-UGC ^a^*^,*b*^, *trnC-GCA*, *trnD-GUC*, *trnE-UUC*, *trnF-GAA*, *trnfM-CAU*, *trnG-GCC ^a^*, *trnH-GUG*, *trnI-CAU ^b^*, *trnI-GAU ^a^*^,*b*^, *trnK-UUU ^a^*, *trnL-CAA ^b^*, *trnL-UAA ^a^*, *trnL-UAG*, *trnM-CAU*, *trnN-GUU ^b^*, *trnP-UGG*, *trnQ-UUG*, *trnR-ACG ^b^*, *trnR-UCU*, *trnS-GCU*, *trnS-GGA*, *trnS-UGA*, *trnT-GGU*, *trnT-UGU*, *trnV-GAC ^b^*, *trnV-UAC ^a^*, *trnW-CCA*, *trnY-GUA*
	Small ribosomal subunit	*rps2*, *rps3*, *rps4*, *rps7* ^b^, *rps8*, *rps11*, *rps12 ^a^*^,b^, *rps14*, *rps15*, *rps16* ^a^, *rps18*, *rps19 ^b^*
	Large ribosomal subunit	*rpl2* ^a,b^, *rpl14*, *rpl16* ^a^, *rpl20*, *rpl22*, *rpl23* ^b^, *rpl32*, *rpl33*, *rpl36*
	RNA polymerase	*rpoA*, *rpoB*, *rpoC1* ^a^, *rpoC2*
Photosynthesis	NADH-dehydrogenase	*ndhA* ^a^, *ndhB* ^a,b^, *ndhC*, *ndhD*, *ndhE*, *ndhF*, *ndhG*, *ndhH*, *ndhI*, *ndhJ*, *ndhK*
	Photosystem Ⅰ	*psaA*, *psaB*, *psaC*, *psaI*, *psaJ*
	Photosystem Ⅱ	*psbA*, *psbB*, *psbC*, *psbD*, *psbE*, *psbF*, *psbH*, *psbI*, *psbJ*, *psbK*, *psbL*, *psbM*, *psbN*, *psbT*, *psbZ*
	RuBisCO ^c^	*rbcL*
	Cytochrome b/f complex	*petA*, *petB ^a^*, *petD ^a^*, *petG*, *petL*, *petN*
	ATP synthase	*atpA*, *atpB*, *atpE*, *atpF* ^a^, *atpH*, *atpI*
Other	Acetyl-CoA carboxylase	*accD*
	Cytochrome c biogenesis	*ccsA*
	Maturase	*matK*
	Envelope membrane protein	*cemA*
Unknown	Conserved hypothetical reading frames	*ycf1* ^b^, *ycf2* ^b^, *ycf3* ^a^, *ycf4*

^a^ Intron-containing gene; ^b^ gene located in IR regions. The *rps12* gene was divided: 5′-*rps12* was located in the LSC region, whereas 3′-*rps12* was located in the IR region; ^c^ RuBisCo, ribulose-1,5-bisphosphate carboxylase/oxygenase.

## Data Availability

The three chloroplast genome sequences are available in GenBank at the National Center for Biotechnology Information (NCBI) under accession numbers KY101151, KY101154, and KY101150 for *P. armeniaca*, *P. sibirica*, and kernel consumption apricot, respectively.

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
