# Peer review of "Sequencing and Phylogenetic Analysis of the Chloroplast Genome of Three Apricot Species"

_genes, 2023, doi:10.3390/genes14101959_

Round 1
Reviewer 1 Report
In my opinion, the article sent for review is important for the development of modern plant genetics, because the authors sequenced, characterized, and phylogenetically analyzed the complete chloroplast genome of Prunus armeniaca and P. sibirica. Moreover, the authors attempted to discuss the obtained results in an evolutionary context. However, They failed to refer the obtained results to the traditional taxonomy. I suggest adding a short paragraph to the discussion chapter, in which the authors will explain what new contributions their research has made in the context of existing knowledge. Do (traditional) morphological analyses confirm the data presented in Figure 6 or do they give a different result? This manuscript generally is well-written, the methodology and conclusions are scientifically sound, and the investigations are extensive. The overall presentation of the data and results is clear and attractive. The conclusions are consistent with the evidence and arguments and address the main questions posed.
Specific comments:
Introduction. The first paragraph needs to be thoroughly rewritten., is not flowing and contains little important information. In the second sentence: “Ten species of apricot have been identified and they are widely distributed worldwide”- please provide details and citations.
Page 7, the Latin names P. sibirica and P. armeniaca should be written in italics
The manuscript should be thoroughly checked for English language and writing errors.
Reviewer 2 Report
I was pleased to review the manuscript entitled: Sequencing and Phylogenetic Analysis of the Chloroplast Genome of Three Apricot Species.
Overall, the manuscript is well written, but minor changes are needed before final publication.
All suggestions are highlighted directly in the text.

Small spelling errors.
Round 2
Reviewer 1 Report
Compared to the original version, the authors have significantly improved the manuscript. This is a much better version of the manuscript than the previous submission. I have no further comments. My recommendation is for the publication of this article.